# Genome-wide identification and characterization of m6A regulatory genes in Soybean: Insights into evolution, miRNA interactions, and stress responses

**Sabrina Bintay Sayed, Md. Afser Rabbi, Joy Prokash Debnath, Kabir Hossen, Ajit Ghosh\***

Department of Biochemistry and Molecular Biology, Shahjalal University of Science and Technology, Sylhet, Bangladesh

\* aghosh-bmb@sust.edu

## Abstract

N6-methyladenosine (m6A) is one of the most prevalent mRNA modifications in eukaryotes, playing a crucial role in plant development and stress responses. The m6A modification is regulated by three key components: writers (which add methyl groups), erasers (which remove them), and readers (which interpret the modification). Despite its significance, the role of m6A regulatory genes in plants, particularly in soybean, remains largely unexplored. This study identified 42 m6A regulatory genes in soybean through a comprehensive genome-wide analysis. Structural analysis revealed diverse gene architectures and functional variations across different sub-groups. A total of 18 gene duplication events were identified, predominantly evolving under purifying selection. Network analysis and hub gene identification suggested weaker interactions among eraser proteins. Moreover, interaction analysis between miRNAs and soybean m6A regulatory genes indicated a stronger miRNA-mediated regulation of writer components compared to erasers and readers. Expression profiling under various stress conditions highlighted distinct regulatory patterns, where *GmECT9*, *GmECT13*, and *GmECT17* were highly expressed in roots and nodules, while *GmMTB2* and *GmALKBH9B2* exhibited maximum upregulation and downregulation, respectively, under combined water deficit and heat stress. Additionally, in the case of mosaic virus infection, *GmALKBH9B4* showed significant downregulation, whereas *GmMTB1*, *GmMTB2*, and *GmALKBH9B1* were upregulated. This study provides a foundational framework for understanding m6A regulatory genes in soybean. The genome-wide insights presented here contribute to a deeper understanding of the molecular mechanisms governing m6A regulation and its potential implications for stress responses and plant development.

**Data availability statement:** All relevant data are within the manuscript and its Supporting Information files.

**Funding:** The author(s) received no specific funding for this work.

**Competing interests:** The authors have declared that no competing interests exist.

**Abbreviations:** amino acid; bp: base pair; CDS: Coding DNA sequence; KDa: kilo Dalton; pI: Iso electric point; m6A: N6-methyladenosine; LFC: Log Fold Change; CC: cellular component; BP: biological process; MF: molecular function; FDR: False discovery rate; FPKM: fragments per kilobase of exon per million; CRE: cis-acting regulatory element; PPI: protein-protein interaction; EN: emerging nodules; MN: mature nodules; ELR: emerging lateral roots: YLR: young lateral roots; SMV: soybean mosaic virus; 3D: three dimensions; Ka: Nonsynonymous rate; Ks: Synonymous rate; Mya: Million years ago.

## Introduction

Plants have continuously suffered from various biotic and abiotic stressors such as pathogens, nutritional imbalance, salt, light intensity, and drought. To cope with these environmental variations, plants must evolve multiple mechanisms [1]. Epigenetic modification is one of them, which is associated with gene expression regulation under stress [2]. Epigenetics refers to stimuli-triggered changes in gene expression without altering the underlying DNA sequence that can be passed to offspring [3]. This process is mainly involved in DNA/RNA methylation, histone modifications, chromatin remodeling, and noncoding RNAs.

To date, RNA modification is an emerging major epigenetic modification in the plant life cycle and plays a critical role in the regulation of gene expression, both at the transcriptional and post-transcriptional levels [4]. Like DNA and protein, RNA goes through 160 types of post-translational modifications, where mRNA modification is found to be less common in comparison to tRNA and rRNA. Among mRNA modifications, N6-methyl-adenosine (m6A) is one of the most abundant, inner, dynamic, and reversible posttranscriptional modifications in eukaryotes and serves as a novel epigenetic marker that is involved in various biological processes [5]. Furthermore, three enzyme complexes, namely methyltransferases (writers), demethylases (erasers), and m6A-binding proteins (readers), are found to be involved in introducing, deleting, and interpreting specific methylation marks on mRNAs, respectively [6].

The writer complex was first discovered in mammals that included methyltransferase-like 3 (METTL3) [7], methyltransferase-like 14 (METTL14) [8], wilms tumor 1-associated protein (WTAP) [9], RNA binding motif protein 15/15B (RBM15/RBM15B) [10], vir-like m6A methyltransferase associated (VIRMA or KIAA1429) [11], zinc finger CCCH-type containing 13 (ZC3H13) [12], methyltransferase-like 16 (METTL16) [13], methyltransferase-like 4 (METTL4) [14], methyltransferase-like 5 (METTL5) [15], and zinc finger CCHC-type containing 4 (ZCCHC4) [16]. The orthologs of these complexes have also been identified in *Arabidopsis,* such as MTA (ortholog of METTL3), MTB (ortholog of METTL14), VIR (ortholog of VIRMA), and FIP37 (ortholog of WTAP) [17–20]. Moreover, E3 ubiquitin ligase HAKAI, an additional m6A writer component, has been identified in *Arabidopsis* [21]. METTL3 (S-adenosyl-methionine-binding protein) is the key catalytic component of the m6A methyltransferase complex. METTL14 and WTAP, making complex with METTL3, are found to be involved in M6A installation and modification, respectively [8]. Besides this, VIRMA is a regulatory subunit of m6A methyltransferase, recruiting the m6A complex to the special RNA site and facilitating m6A installation [11].

Conversely, the m6A demethylase erases the methyl group from the mRNA, indicating the reversibility of m6A modification. FTO (fat mass and obesity-associated protein) is the first eraser complex discovered in mammals that facilitates in restoration of the methylated base to the adenine base [22]. Subsequently, ALKBH5 (alkB homolog 5), the second mammalian m6A demethylase facilitating the removal of m6A, potentially affects different subsets of target mRNAs [23]. *Arabidopsis* contains several putative m6A eraser ALKBH family proteins [20]. ALKBH9B and ALKBH10B

are the only two eraser proteins found in *Arabidopsis* that have been functionally investigated concerning viral infection and floral transition [24].

While writers and erasers are actively engaged in their dynamic action, the reader family regulates mRNA nuclear export, splicing, degradation, translation, and stability. The YT521-B homology (YTH) domain family is the first identified mammalian reader complex, which includes YTHDF1, YTHDF2, YTHDF3, YTHDC1, and YTHDC2 [25]. Different reader proteins have different functions on m6A-modified RNAs. For example, YTHDF2 regulates the degradation [26], YTHDF1 modulates translation [27], FMRP enhances nuclear export and stability [28], and YTHDC1 modulates nuclear export and splicing [29]. Furthermore, YTHDC2 and Eukaryotic initiation factor 3 (eIF3) are also crucial in mRNA translation [27,30]. Notably, three evolutionarily conserved C-terminal regions (ECT) family proteins have recently been functionally characterized in Arabidopsis as YTHD homologs [31,32].

The m6A modification is a crucial regulator performing three main functions: mRNA processing, plant growth and development, and stress response [33]. Researchers suggest that m6A regulatory writer proteins are essential for shoot and root growth, leaf, cotyledon, and floral development [18]. Partial loss of FIP37 causes huge over-proliferation of shoot meristems. Moreover, depletion of FIP37 results in embryo lethality [17]. The m6A modification-associated genes are expressed differentially in response to biotic and abiotic stresses, including drought [34], salt [35], cold [36], and bacterial infections [37]. Nevertheless, m6A modifications have been studied in limited plant species, highlighting the need for future investigation in other plants.

Soybean (*Glycine max*), a Leguminosae family member, has become one of the main economic oilseed beans [38]. Currently, researchers are shedding light on pinpointing stress-responsive writer as well as eraser genes in soybean [39,40]. Nevertheless, comprehensive or systematic analysis of all the m6A gene families remains scarce. Therefore, we were encouraged to conduct genome-wide identification of writers, erasers, and readers genes of the m6A pathway in soybean. A total of 12 writers, 11 erasers, and 19 readers were identified. A comprehensive analysis of the m6A regulatory genes in soybeans, including classification, evolution, gene structure, and potential interaction network, was performed. Expression patterns in response to a range of biotic and abiotic stresses were also studied.

## Materials and methods

### Genome-wide identification of the m6A gene family in Soybean

To identify m6A components and their protein families in soybeans, the amino acid sequences of m6A-related proteins reported in *Arabidopsis thaliana* [41], including writers, erasers, and readers, were used as queries to perform BLASTP against the soybean genomic sequences in the plant genomics resource database phytozome [42]. Putative genes and corresponding protein sequences were retrieved from the same database, and the presence of conserved domains was confirmed by NCBI Conserved Domain (https://www.ncbi.nlm.nih.gov/Structure/cdd/wrpsb.cgi) and Pfam (http://pfam.xfam.org/). The identity of the "writer" family was confirmed by the presence of the MT-A70 domain (PF05063), Wtap domain (PF17098), and Vir-N domain (PF15912). The presence of 2-OG Fe (II) oxygenase superfamily (PF13532) and YTH domain (PF04146) confirmed the identity of the "eraser" and "reader" families, respectively. Candidates without the m6A conserved and typical domains were not further considered. The prefix "Gm" for *Glycine max* was added to all members, followed by subclass IDs (MT, FIP37, VIRILIZER, ECT, CPSF, and ALKBH). Chromosomal location, strand position, CDS coordinate (5' to 3'), gene length, CDS length, and protein length were retrieved from the Phytozome database [42]. Furthermore, the physicochemical properties of the identified proteins, such as molecular weight and theoretical isoelectric point, were calculated using the ExPASy-ProtParam tool [43]. Subcellular localization of proteins was predicted using BUSCA [44].

### Chromosomal localization and gene duplication events

The GFF (General Feature Format) file of soybean was retrieved from the phytozome database [42]. Chromosomal locations were visualized by TBtool [45]. Using the Plant Genome Duplication Database (PGDD, http://pdgd.njau.edu.cn:8080/)

[46], paralogous genes were found by calculating the number of duplication events among soybean genes. From the same database, the synonymous substitution rate (Ks) and non-synonymous substitution rate (Ka) were obtained. Using the Ka/Ks ratio, the selection pressure of duplicated genes was computed. Ka/Ks < 1 denotes purifying selection, Ka/Ks = 1 implies neutral drifting, and Ka/Ks > 1 signifies positive or Darwinian selection for a pair of genes [47]. Using $T = Ks/ (2 \times 6.1 \times 10^{-9}) \times 10^{-6}$ Mya formula, the approximate date (Mya, million years ago) of each duplication event was estimated [48].

## Gene structure, conserved domain, and conserved motif analysis

Genomic and CDS sequences were retrieved from the phytozome database [42]. Subsequently, the physical mapping of the exon-intron sequence was plotted using GSDS (Gene Structure Display Server 2.0, http://gsds.gao-lab.org/). The conserved domains of m6A regulatory protein sequences were analyzed in the Batch CD-Search program [49], and then the output file was submitted to TBtools for visualizing analyses. The conserved motifs were analyzed using TBtools with a maximum number of motifs of 10 [45].

## Phylogenetic analysis

Protein sequences of writer, eraser, and reader of 13 species, namely *Linum usitatissimum*, *Acorus americanus*, *Anacardium occidentale*, *Aquilegia coerulea*, *Brachypodium hybridum*, *Coffea arabica*, *Gossypium barbadense*, *Oryza sativa*, *Panicum hallii*, *Solanum lycopersicum*, *Triticum aestivum*, *Zea mays*, and *Porphyra umbilicalis* were retrieved from the phytozome database. Multiple sequence alignment was performed using the MUSCLE algorithm [50], and phylogenetic trees for the protein families were constructed using Molecular Evolutionary Genetics Analysis (MEGA) 11 software [51]. The maximum likelihood method was employed with node reliability assessed through bootstrap analysis based on 1000 replicates. The resulting phylogenetic trees were further refined and graphically represented using the Interactive Tree of Life (iTOL) platform (https://itol.embl.de).

## Promoter analysis in Soybean m6A regulatory genes

The 1000 bp upstream sequences from the transcription start site were collected and analyzed through the PlantCare database [52] for cis-acting element analysis. The presence of identified cis-acting elements on the putative promoter of every gene had been depicted using TBtool [45].

## Functional enrichment visualization of m6A regulatory genes

Gene ontology enrichment in three categories, such as cellular component (CC), biological process (BP), and molecular function (MF), was executed using the STRING v12 database (https://string-db.org/). FDR (False discovery rate) =<0.05 was set. The number of terms was shown as 10. The results were shown as a bar chart and a bubble chart.

## Protein-protein interactions, cluster prediction, and hub gene identifications

The STRING v12 database (https://string-db.org/) was used to construct the protein-protein interaction network among m6A regulatory genes in soybeans. The output file from STRING was visualized by Cytoscape software [53]. The network's co-expressing network clusters (highly linked areas) were examined using the Molecular Complex Detection (MCODE) (v2.0.2) plugin [54]. Hub genes were recognized using the cytoHubba plugin [55] in Cytoscape [53] with the shortest path and ranked by degree.

## Structure construction by homology modeling

The homology modeling approach was used to predict the three-dimensional (3D) structures of the m6A regulatory proteins. The amino acid sequence was queried against the SWISS-MODEL server (https://swissmodel.expasy.org/) [56] to

search for templates, and the best templates with a similar amino acid sequence and known three-dimensional structures were used to build the models. These proteins' structures were examined with the UCSF ChimeraX [57] visualization tools.

### microRNA targets prediction

Mature miRNAs were downloaded from the miRBase database [58]. The psRNAT-target database [59] was used to search the regulatory relationship between miRNAs and soybean m6A regulatory genes, with an Expectation threshold of < 5 and other parameters at their default values. Cytoscape [53] was used to visualize the interaction network.

### Expression analysis of m6A regulatory genes

RNAseq datasets (GSE137263 [60], GSE129509 [61], and GSE186317 [62]) were downloaded from the GEO database (https://www.ncbi.nlm.nih.gov/geo/) of the NCBI (https://www.ncbi.nlm.nih.gov/) to observe the expression pattern of m6A genes in different organs, biotic, and abiotic stress. Normalized FPKM (fragments per kilobase of exon per million) data were used to observe expression levels in emerging nodules, mature nodules, emerging lateral roots, and mature lateral roots. To identify the expression patterns in abiotic stress, including salt stress, heat stress, dehydration, water deficit, and combined water deficit and heat stress, Log Fold Change (LFC) values were extracted using the R package DESeq2 [63]. The expression pattern of m6A regulatory genes in soybean mosaic virus (SMV) infection was also observed. The heat-maps were plotted using TBtool with normalized FPKM (roots and nodules) and LFC values (abiotic and biotic stress).

## Results

### Genome-wide identification and characterization of m6A regulatory genes in Soybean

After a comprehensive screening of the soybean genome, 12 m6A writers, 11 m6A erasers, and 19 m6A readers were identified. All m6A regulatory genes were named according to their subclass identifier. The amino acid sequence length, relative molecular weights (MWs), isoelectric points (pI), subcellular localization, and gravy were calculated (Table 1). It was found that the soybean writer's CDS length ranged from 624 bp to 6561 bp. *GmVIRILIZER2* is the biggest member of the writer family and has a polypeptide length of 2187 aa and a molecular weight of 240.88 kDa. Meanwhile, GmFIP37c is the smallest, having a polypeptide length of 208 aa and a molecular weight of 23.13 kDa. The pI values of the writer family differed widely, ranging from 5.04 (GmFIP37a) to 8.81 (GmMTC2). Among the 12 writer members, ten exhibit acidic pI, while the remaining two show basic features. Likewise, the CDS length of the m6A eraser varies from 957 bp to 2052 bp. GmALKBH9B3 is the longest eraser protein with a 684 aa long and a molecular weight of 73.46 kDa. On the other hand, the smallest one, GmALKBH10B4, is 957 aa long with a molecular weight of 35.39 kDa. In terms of pI, eight of eleven erasers have acidic pI, whereas three others possess basic pI. The predicted CDS length of the m6A reader also varies from 387 bp (*GmECT15*) to 2127 bp (*GmECT9*). The largest one (GmECT9) is 77.17 kDa in size and 709 aa long. The smallest one (GmECT15) is 129 aa long with a molecular weight of 14.59 kDa. In terms of localization, most m6A regulatory proteins were in the nucleus, and only several proteins were in the plasma membrane (GmECT13, GmALKBH10B5, GmECT9) and chloroplast (GmMTC2, GmECT6). The Grand Average of Hydropathicity (GRAVY) values for all m6A proteins are less than zero, indicating that all soybean m6A proteins are hydrophilic.

### Chromosomal localization and gene duplication

The identified putative 42 genes have been distributed throughout the 17 chromosomes of soybeans (Table 1 and S1 Fig). The distribution of genes across the chromosomes can be summarized as follows: Chromosomes 17 and 8 each have the highest count, with four genes. Following closely, chromosomes 16, 10, 14, 5, and 2 each have three genes. Chromosomes 20 and 19 have two genes each. Finally, chromosomes 4, 6, 3, 1, 12, 9, and 15 each have a single gene in their

**Table 1. List of identified RNA m6A genes in *Glycine max* along with their detailed information and subcellular localization.**

| Type | Sl no | Gene name | Transcript | | | | Protein | | | | | |
|------|-------|-----------|------------|------------------|--------|-------------|-------------|-------------|------|------------------|--------|
| | | | Locus ID | Location (Start..End) | Strand | CDS (bp) | Length (aa) | MW (KDa) | pl | Localization | Gravy |
| Writer | 1 | GmMTA1 | Glyma.07G067100.1 | 6047500..6052182 | Reverse | 2289 | 763 | 84.61 | 6.22 | Nucleus | −0.491 |
| | 2 | GmMTA2 | Glyma.16G033100.1 | 3127715..3133331 | Reverse | 2286 | 762 | 84.26 | 5.95 | Nucleus | −0.465 |
| | 3 | GmMTB1 | Glyma.10G232300.1 | 46180119..46186843 | forward | 3309 | 1103 | 121.73 | 6.65 | Nucleus | −1.169 |
| | | | Glyma.10G232300.2 | 46180119..46186843 | forward | 3309 | 1103 | 121.73 | 6.65 | Nucleus | −1.169 |
| | 4 | GmMTB2 | Glyma.20G161800.1 | 39955682..39962442 | Reverse | 3297 | 1099 | 121.47 | 6.76 | Nucleus | −1.141 |
| | | | Glyma.20G161800.2 | 39955682..39962442 | Reverse | 3297 | 1099 | 121.47 | 6.76 | Nucleus | −1.141 |
| | | | Glyma.20G161800.3 | 39955682..39962442 | Reverse | 3297 | 1099 | 121.47 | 6.76 | Nucleus | −1.141 |
| | 5 | GmMTC1 | Glyma.14G077000.1 | 6455980..6460708 | Reverse | 1287 | 429 | 48.878 | 7.14 | Nucleus | −0.425 |
| | | | Glyma.14G077000.2 | 6455980..6460708 | Reverse | 1266 | 422 | 48.062 | 6.83 | Nucleus | −0.418 |
| | | | Glyma.14G077000.3 | 6455980..6460708 | Reverse | 1212 | 404 | 46.025 | 6.8 | Nucleus | −0.476 |
| | 6 | GmMTC2 | Glyma.17G248800.1 | 40410224..40414094 | forward | 1056 | 352 | 39.89 | 8.81 | Chloroplast | −0.228 |
| | 7 | GmFIP37a | Glyma.17G086600.1 | 6689132..6695722 | Reverse | 1032 | 344 | 38.75 | 5.04 | Nucleus | −0.862 |
| | | | Glyma.17G086600.2 | 6689132..6695722 | Reverse | 1023 | 341 | 38.42 | 5.08 | Nucleus | −0.871 |
| | 8 | GmFIP37b | Glyma.04G186400.1 | 45656967..45664438 | forward | 1065 | 355 | 40.34 | 5.75 | Nucleus | −0.961 |
| | | | Glyma.04G186400.2 | 45656967..45664438 | forward | 888 | 269 | 34.02 | 6.7 | Nucleus | −0.92 |
| | | | Glyma.04G186400.3 | 45656967..45664438 | forward | 1065 | 355 | 40.34 | 5.75 | Nucleus | −0.961 |
| | 9 | GmFIP37c | Glyma.05G040200.1 | 3582201..3589663 | forward | 1020 | 340 | 38.34 | 5.12 | Nucleus | −0.879 |
| | | | Glyma.05G040200.2 | 3582201..3589663 | forward | 762 | 254 | 28.30 | 5.6 | Nucleus | −0.766 |
| | | | Glyma.05G040200.3 | 3582201..3589663 | forward | 624 | 208 | 23.13 | 5.9 | Nucleus | −0.705 |
| | 10 | GmFIP37d | Glyma.06G179400.1 | 15233147..15241225 | Reverse | 1032 | 344 | 39.17 | 5.59 | Nucleus | −0.998 |
| | | | Glyma.06G179400.2 | 15233147..15241225 | Reverse | 855 | 285 | 32.77 | 5.6 | Nucleus | −0.913 |
| | | | Glyma.06G179400.3 | 15233147..15241225 | Reverse | 900 | 300 | 33.93 | 5.7 | Nucleus | −0.764 |
| | 11 | GmVIRILIZER1 | Glyma.10G082100.1 | 9934485..9960586 | Reverse | 6525 | 2175 | 239.33 | 5.36 | Nucleus | −0.125 |
| | 12 | GmVIRILIZER2 | Glyma.02G195600.1 | 36909269..36924536 | forward | 6561 | 2187 | 240.88 | 5.34 | Nucleus | −0.145 |
| Eraser | 1 | GmALKBH10B1 | Glyma.10G023900.1 | 2089240..2094838 | reverse | 1536 | 512 | 56.08 | 5.71 | Nucleus | −0.35 |
| | | | Glyma.10G023900.2 | 2089240..2094838 | reverse | 1536 | 512 | 40.62 | 4.95 | Nucleus | −0.365 |
| | 2 | GmALKBH10B2 | Glyma.07G175300.1 | 31882295..31888358 | reverse | 1995 | 665 | 71.74 | 6.61 | Nucleus | −0.444 |
| | 3 | GmALKBH10B3 | Glyma.02G149900.1 | 15416607..15422632 | forward | 1545 | 515 | 56.37 | 5.95 | Nucleus | −0.323 |
| | 4 | GmALKBH10B4 | Glyma.19G152900.1 | 41308826..41318638 | reverse | 1560 | 520 | 57.23 | 5.89 | Nucleus | −0.371 |
| | | | Glyma.19G152900.2 | 41308826..41318638 | reverse | 1557 | 519 | 57.09 | 5.89 | Nucleus | −0.365 |
| | | | Glyma.19G152900.3 | 41308826..41318638 | reverse | 1557 | 519 | 57.09 | 5.89 | Nucleus | −0.365 |
| | | | Glyma.19G152900.4 | 41308826..41318638 | reverse | 957 | 319 | 35.39 | 5.11 | Plasma membrane | −0.456 |
| | | | Glyma.19G152900.5 | 41308826..41318638 | reverse | 1107 | 369 | 40.90 | 5.14 | Plasma membrane | −0.405 |
| | | | Glyma.19G152900.6 | 41308826..41318639 | reverse | 1089 | 363 | 40.24 | 8.66 | Nucleus | −0.316 |
| | | | Glyma.19G152900.7 | 41308826..41318640 | reverse | 1107 | 369 | 40.37 | 8.66 | Nucleus | −0.325 |
| | 5 | GmALKBH10B5 | Glyma.03G149900.1 | 36498290..36505316 | reverse | 1566 | 522 | 57.58 | 5.88 | Plasma membrane | −0.385 |
| | | | Glyma.03G149900.2 | 36498290..36505317 | reverse | 1563 | 521 | 57.45 | 5.88 | Plasma membrane | −0.379 |
| | 6 | GmALKBH10B6 | Glyma.20G012100.1 | 1010564..1017683 | reverse | 1881 | 627 | 68.04 | 6.73 | Nucleus | −0.419 |
| | | | Glyma.05G138600.2 | 33089658..33095963 | reverse | 1926 | 642 | 69.46 | 7.36 | Nucleus | −0.517 |
| | 7 | GmALKBH10B7 | Glyma.14G106000.1 | 10920596..10924875 | reverse | 1572 | 524 | 58.56 | 6.22 | Nucleus | −0.732 |
| | | | Glyma.14G106000.2 | 10920596..10924876 | reverse | 1569 | 523 | 58.42 | 6.11 | Nucleus | −0.725 |
| | 8 | GmALKBH9B1 | Glyma.17G220300.1 | 37245512..37249806 | forward | 1563 | 521 | 58.29 | 6.3 | Nucleus | −0.752 |
| | 9 | GmALKBH9B2 | Glyma.05G138600.1 | 33089658..33095962 | reverse | 2046 | 682 | 73.69 | 6.79 | Nucleus | −0.535 |
| | 10 | GmALKBH9B3 | Glyma.08G093800.1 | 7102856..7109102 | reverse | 2052 | 684 | 73.46 | 7.12 | Nucleus | −0.503 |
| | 11 | GmALKBH9B4 | Glyma.08G186500.1 | 14947244..14951034 | reverse | 1272 | 424 | 47.99 | 6.29 | Nucleus | −0.544 |

*(Continued)*

| Type | Sl no | Gene name | Transcrript | | | | Protein | | | | |
|------|-------|-----------|-------------|--------------------|--------|-------------|-------------|------------|------|----------------------|--------|
| | | | Locus ID | Location (Start..End) | Strand | CDS (bp) | Length (aa) | MW (KDa) | pI | Localization | Gravy |
| **Reader** | 1 | GmECT1 | Glyma.17G038400.1 | 2836786..2841979 | reverse | 2085 | 695 | 76.50 | 5.75 | Nucleus | −0.547 |
| | | | Glyma.17G038400.2 | 2836786..2841979 | reverse | 2085 | 695 | 76.50 | 5.75 | Nucleus | −0.547 |
| | 2 | GmECT2 | Glyma.09G031200.1 | 2566173..2573725 | forward | 1335 | 445 | 50.01 | 6.14 | Nucleus | −0.679 |
| | 3 | GmECT3 | Glyma.07G233400.1 | 41461525..41466672 | reverse | 2088 | 696 | 76.82 | 6.01 | Nucleus | −0.516 |
| | 4 | GmECT4 | Glyma.07G000900.1 | 99314..107313 | forward | 1944 | 648 | 70.88 | 5.42 | Extra cellular | −0.572 |
| | | | Glyma.07G000900.2 | 99314..107313 | forward | 1914 | 638 | 69.64 | 5.28 | Nucleus | −0.646 |
| | 5 | GmECT5 | Glyma.15G265400.1 | 50001901..50006882 | reverse | 1734 | 578 | 63.71 | 6.9 | Nucleus | −0.665 |
| | | | Glyma.15G265400.2 | 50001901..50006882 | reverse | 1728 | 576 | 63.48 | 6.9 | Nucleus | −0.668 |
| | | | Glyma.15G265400.3 | 50001901..50006882 | reverse | 1422 | 474 | 51.66 | 8.6 | Nucleus | −0.444 |
| | 6 | GmECT6 | Glyma.15G136400.1 | 11025561..11030449 | forward | 2106 | 702 | 76.84 | 5.94 | Chloroplast | −0.473 |
| | | | Glyma.15G136400.2 | 11025561..11030449 | forward | 1971 | 657 | 72.33 | 5.97 | Nucleus | −0.561 |
| | | | Glyma.15G136400.3 | 11025561..11030449 | forward | 1971 | 621 | 68.22 | 6.58 | Nucleus | −0.546 |
| | 7 | GmECT7 | Glyma.02G072000.1 | 6320072..6325195 | reverse | 1893 | 631 | 68.63 | 7.6 | Nucleus | −0.703 |
| | | | Glyma.02G072000.2 | 6320072..6325195 | reverse | 1887 | 629 | 68.46 | 7.95 | Nucleus | −0.707 |
| | | | Glyma.02G072000.3 | 6320072..6325195 | reverse | 1617 | 539 | 59.07 | 7.61 | Nucleus | −0.732 |
| | 8 | GmECT8 | Glyma.19G110800.1 | 36458959..36463094 | reverse | 1980 | 660 | 71.74 | 8.45 | Nucleus | −0.677 |
| | | | Glyma.19G110800.2 | 36458959..36463094 | reverse | 1476 | 492 | 53.89 | 8.59 | Nucleus | −0.69 |
| | | | Glyma.19G110800.3 | 36458959..36463094 | reverse | 1728 | 576 | 62.75 | 8.61 | Nucleus | −0.681 |
| | | | Glyma.19G110800.4 | 36458959..36463094 | reverse | 1728 | 576 | 62.88 | 8.39 | Extra cellular | −0.685 |
| | | | Glyma.19G110800.5 | 36458959..36463094 | reverse | 1695 | 565 | 61.52 | 8.63 | Nucleus | −0.618 |
| | 9 | GmECT9 | Glyma.05G166600.1 | 35712788..35717009 | reverse | 2127 | 709 | 77.17 | 6.19 | Plasma membrane | −0.732 |
| | | | Glyma.05G166600.2 | 35712788..35717009 | reverse | 2040 | 680 | 74.18 | 5.97 | Plasma membrane | −0.726 |
| | | | Glyma.05G166600.3 | 35712788..35717010 | reverse | 2022 | 674 | 73.63 | 5.87 | Plasma membrane | −0.72 |
| | 10 | GmECT10 | Glyma.11G027800.1 | 2000191..2004609 | forward | 807 | 269 | 29.84 | 6.83 | Nucleus | −0.551 |
| | 11 | GmECT11 | Glyma.08G226100.1 | 18339383..18347070 | reverse | 1977 | 659 | 72.05 | 5.51 | Chloroplast | −0.625 |
| | | | Glyma.08G226100.2 | 18339383..18347071 | reverse | 1878 | 626 | 68.55 | 5.34 | Nucleus | −0.667 |
| | | | Glyma.08G226100.3 | 18339383..18347072 | reverse | 1911 | 637 | 69.66 | 5.29 | Nucleus | −0.683 |
| | | | Glyma.08G226100.4 | 18339383..18347073 | reverse | 1911 | 637 | 69.66 | 5.29 | Nucleus | −0.683 |
| | | | Glyma.08G226100.5 | 18339383..18347074 | reverse | 1878 | 626 | 68.55 | 5.34 | Nucleus | −0.667 |
| | 12 | GmECT12 | Glyma.08G162200.1 | 12635477..12640656 | forward | 1740 | 580 | 63.78 | 6.58 | Nucleus | −0.701 |
| | | | Glyma.08G162200.2 | 12635477..12640657 | forward | 1698 | 566 | 62.34 | 8.27 | Nucleus | −0.7 |
| | 13 | GmECT13 | Glyma.08G124600.1 | 9570497..9574730 | reverse | 2124 | 708 | 77.07 | 6.35 | Plasma membrane | −0.743 |
| | | | Glyma.08G124600.2 | 9570497..9574731 | reverse | 2037 | 679 | 74.09 | 6.12 | Plasma membrane | −0.74 |
| | 14 | GmECT14 | Glyma.01G214100.1 | 54514431..54518897 | reverse | 1191 | 397 | 44.44 | 5.59 | Nucleus | −0.704 |
| | | | Glyma.01G214100.2 | 54514431..54518898 | reverse | 1065 | 355 | 39.64 | 5.52 | Nucleus | −0.608 |
| | | | Glyma.01G214100.3 | 54514431..54518899 | reverse | 1065 | 355 | 39.64 | 5.52 | Nucleus | −0.608 |
| | | | Glyma.01G214100.4 | 54514431..54518900 | reverse | 1074 | 358 | 39.94 | 5.43 | Nucleus | −0.617 |
| | | | Glyma.01G214100.5 | 54514431..54518901 | reverse | 1074 | 358 | 39.94 | 5.43 | Nucleus | −0.617 |
| | | | Glyma.01G214100.6 | 54514431..54518902 | reverse | 1182 | 394 | 44.14 | 5.67 | Nucleus | −0.696 |
| | | | Glyma.01G214100.7 | 54514431..54518903 | reverse | 1182 | 394 | 44.14 | 5.67 | Nucleus | −0.696 |
| | | | Glyma.01G214100.8 | 54514431..54518904 | reverse | | 397 | 44.45 | 5.59 | Nucleus | −0.704 |

*(Continued)*

| Type | Sl no | Gene name | Transcrript | | | | | Protein | | | | |
|------|-------|-----------|-------------|---|---|---|---|---------|---|---|---|---|
| | | | Locus ID | Location (Start..End) | Strand | CDS (bp) | Length (aa) | MW (KDa) | pI | Localization | Gravy |
| | 15 | GmECT15 | Glyma.12G167200.1 | 32158767..32161717 | reverse | 387 | 129 | 14.59 | 8.87 | Nucleus | −0.356 |
| | 16 | GmECT16 | Glyma.16G003200.1 | 172939..178668 | reverse | 1491 | 497 | 56.04 | 9.5 | Nucleus | −0.655 |
| | | | Glyma.16G003200.2 | 172939..178669 | reverse | 1431 | 477 | 53.64 | 9.45 | Nucleus | −0.701 |
| | | | Glyma.16G003200.3 | 172939..178670 | reverse | 1206 | 402 | 43.008 | 9.55 | Chloroplast | −0.634 |
| | | | Glyma.16G003200.4 | 172939..178671 | reverse | 1122 | 374 | 46.06 | 9.65 | Chloroplast | −0.68 |
| | 17 | GmECT17 | Glyma.16G041600.1 | 3904130..3908552 | forward | 1992 | 664 | 72.12 | 7.15 | Nucleus | −0.681 |
| | 18 | GmCPSF30a | Glyma.09G022200.1 | 1768806..1778364 | forward | 2046 | 682 | 74.15 | 6.19 | Nucleus | −0.835 |
| | 19 | GmCPSF30b | Glyma.15G128500.1 | 10243235..10253450 | forward | 2076 | 692 | 75.35 | 6.41 | Nucleus | −0.851 |

**Abbreviations:** CDS, coding DNA Sequence; PP, Polypeptide; MW, Molecular Weight; pI, Isoelectric point; bp, base pair; aa, amino acid; KDa, kilo Dalton.

respective location. Different duplication events of m6A regulatory genes in soybeans were identified through PGDD [46], as gene duplication is an important mechanism for gene family expansion. A total of 18 duplication events were observed (Table 2 and S1 Fig). Six writer's sister gene pairs (*GmCPSF30a-GmCPSF30b*, *GmMTA1-GmMTA2*, *GmMTB1-GmMTB2*, *GmFIP37a-GmFIP37c*, *GmFIP37b-GmFIP37d*, *GmVIRILIZER1-GmVIRILIZER2*) were observed to be duplicated. Alongside, five erasers were discovered to be duplicated: *GmALKBH10B7-GmALKBH9B1*, *GmALKBH10B1-GmALKBH10B3*, *GmALKBH10B4-GmALKBH10B5*, *GmALKBH10B2-GmALKBH10B6*, and *GmALKBH9B2-GmALKBH9B3*. In addition,

**Table 2. List of duplicated RNA m6A genes identified in soybean.**

| Sl no | Locus 1 | Locus 2 | Ka | Ks | Ka/Ks | Duplication time (Mya) | Duplication type | Purifying selection |
|-------|---------|---------|-----|-----|-------|------------------------|------------------|---------------------|
| 1 | *GmECT4* | *GmECT11* | 0.032505615 | 0.098584527 | 0.329723292 | 8.080698901 | Segmental | Yes |
| 2 | *GmECT15* | *GmECT16* | 0.034067736 | 0.189125753 | 0.180132716 | 15.5021109 | Segmental | Yes |
| 3 | *GmECT5* | *GmECT12* | 0.023263591 | 0.126486191 | 0.183921984 | 10.36772059 | Segmental | Yes |
| 4 | *GmECT9* | *GmECT13* | 0.016805682 | 0.15871889 | 0.105883316 | 13.00974509 | Segmental | Yes |
| 5 | *GmECT8* | *GmECT17* | 0.03864886 | 0.142561735 | 0.271102618 | 11.6853881 | Segmental | Yes |
| 6 | *GmECT1* | *GmECT3* | 0.030299181 | 0.069726147 | 0.434545474 | 5.715257924 | Segmental | Yes |
| 7 | *GmVIRILIZER1* | *GmVIRILIZER2* | 0.029112952 | 0.085367773 | 0.341029764 | 6.997358455 | Segmental | Yes |
| 8 | *GmECT10* | *GmECT14* | 0.056837091 | 0.102384016 | 0.555136374 | 8.392132424 | Segmental | Yes |
| 9 | *GmCPSF30a* | *GmCPSF30b* | 0.012173252 | 0.118162892 | 0.103020937 | 9.685482913 | Segmental | Yes |
| 10 | *GmMTA1* | *GmMTA2* | 0.029601946 | 0.097318704 | 0.304175304 | 7.976942961 | Segmental | Yes |
| 11 | *GmMTB1* | *GmMTB2* | 0.01668943 | 0.111476456 | 0.149712602 | 9.137414442 | Segmental | Yes |
| 12 | *GmFIP37a* | *GmFIP37c* | 0.017546492 | 0.124153707 | 0.141328781 | 10.1765334 | Segmental | Yes |
| 13 | *GmFIP37b* | *GmFIP37d* | 0.021262497 | 0.096867569 | 0.219500682 | 7.939964661 | Segmental | Yes |
| 14 | *GmALKBH10B7* | *GmALKBH9B1* | 0.042563592 | 0.146924355 | 0.289697322 | 12.04297989 | Segmental | Yes |
| 15 | *GmALKBH10B1* | *GmALKBH10B3* | 0.020020047 | 0.111401278 | 0.179711108 | 9.131252268 | Segmental | Yes |
| 16 | *GmALKBH10B4* | *GmALKBH10B5* | 0.035220727 | 0.138953453 | 0.253471403 | 11.38962732 | Segmental | Yes |
| 17 | *GmALKBH10B2* | *GmALKBH10B6* | 0.039390877 | 0.119135324 | 0.330639777 | 9.765190516 | Segmental | Yes |
| 18 | *GmALKBH9B2* | *GmALKBH9B3* | 0.036195202 | 0.088239728 | 0.410191674 | 7.232764628 | Segmental | Yes |

sister gene pairs were shown to be duplicated in seven readers (*GmECT4-GmECT11*, *GmECT15-GmECT16*, *GmECT5-GmECT12*, *GmECT9-GmECT13*, *GmECT8-GmECT17*, *GmECT1-GmECT3*, and *GmECT10-GmECT14*). All the duplication events of m6A regulatory genes were segmental duplications (Table 2). To explore the evolutionary history and direction of selection, non-synonymous to synonymous substitution (Ka/Ks) ratios were calculated. Purifying selection, neutral selection, and positive selection were indicated by Ka/Ks ratios below 1, equal to 1, and above 1, respectively. In this study, all analyzed duplication events yielded Ka/Ks ratios that were less than 1, indicating the influence of purifying selection on these genes throughout their evolutionary history. The time of divergence of these gene pairs varied from 5.71 to 15.50 Mya (Table 2).

## Structural analysis of writer, eraser, and reader members

Using amino acid sequences, maximum likelihood phylogenetic trees were built to investigate the structure and sequencing properties of the writer, eraser, and reader genes in soybeans (Fig 1A). All the writer genes, such as *GmMTA*, *GmMTB*, *GmMTC*, *GmVIRILIZER*, and *GmFIP37* clustered according to their group. Likewise, all the *GmALKBH10*B genes clustered together except *GmALKBH10B7*, which formed a cluster with *GmALKBH9B1*. In the reader group, most of the *GmECT* genes clustered in the same clade. However, *GmECT15* and *GmECT16* were placed away from those clusters. Besides, *GmECT10* and *GmECT14* were clustered with *GmCPSF30* genes rather than *GmECT* genes. The

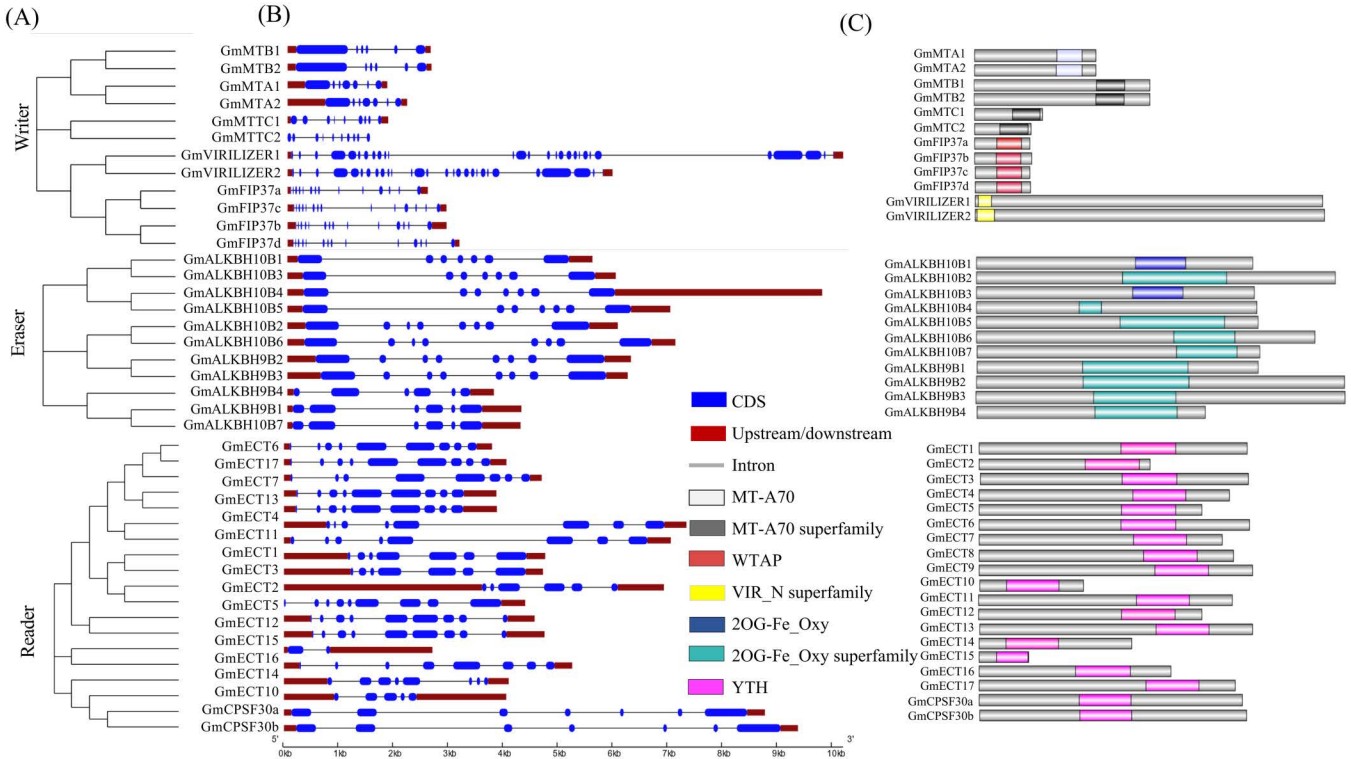

**Fig 1. The evolutionary relationship, gene structure, and the appropriate domain architecture of the m6A-regulating genes in soybean.** A. Using MEGA11, three groups' phylogenetic trees were constructed. B. The gene structures were plotted by using GSDS. The exon-intron length was proportionally shown and could be approximated using the scale below. Exons are shown as blue boxes, upstream/downstream sequences are represented as dark red boxes, and introns are shown as grey lines. C. Functional domains of m6A regulatory genes in soybean. The YTH reader domains were marked in lavender, while the Eraser domain 2OG-Fe (II) oxy and superfamily domains were indicated by sky blue and cyan color, respectively. The MTA-70, MTA-70 superfamily, WTAP, and VIR-N writer domains were visually distinguished by light grey, dark grey, red, and yellow, respectively.

exon-intron distribution indicates the diversity of gene functions. According to the gene structure, the exon numbers of soybean m6A writers, erasers, and readers varied from 6–27, 6–8, and 2–9, respectively (Fig 1B). Amongst the m6A writers, the highest number of exons was observed in *GmVIRILIZER1* (27), and the lowest number was seen in *GmMTB1* (6) and *GmMTB2* (6). In terms of eraser, all the genes contain only 6 to 8 exons, indicating the lowest number of exons among the three m6A regulatory proteins. Within readers, *GmECT15* contains only two exons.

Group-specific conserved domains analysis indicates MT-A70, Wtap, HAKAI, and VIR_N as m6A writers. MT-70 domain was found in MTA, while the MT-70 superfamily was found in both MTB and MTC proteins (Fig 1C). In addition, *GmFIP37a*, *GmFIP37b*, *GmFIP37c*, and *GmFIP37d* were characterized with the WTAP superfamily. The VIR_N super-family was found in *GmVIRILIZER1* and *GmVIRILIZER2*. We found *GmHAKAI* according to phytozome. However, Pfam analysis did not show a HAKAI-related domain for *GmHAKAI*. Therefore, this gene was excluded from downstream analysis. The eraser family is characterized by the 2OG-FeII_Oxy superfamily (Fe (II) dependent Oxygenase superfamily) and 2OG-FeII_Oxy, all the identified putative eraser genes containing these conserved domains (Fig 1C). YTH (YT-521-B-like domain) was used to identify reader proteins, and all the identified readers contained that domain (Fig 1C). In the motif analysis, MTC1 and MTC2 contain only one motif (S2A Fig). Other members of the writer contain 4 to 6 motifs. All the eraser families contain 5–9 conserved motifs, while 2–5 motifs were observed in the reader family (S2B and S2C Fig).

## The phylogenetic classifications of m6A regulatory genes in soybean

Demonstrating phylogenetic relationships is crucial as it gives insight into deducing the gene origins, examining their molecular adaptation, learning about the evolution of morphological traits, and rebuilding demographic variations in recently differentiated species [64]. Phylogenetic trees for m6A writers, erasers, and readers were constructed to study the evolutionary relationships of m6A modifiers. All the writer family, MT, FIP37(WTAP), VIR_N, and HAKAI formed four groups named I.II.III and IV (Fig 2A). The phylogeny revealed that GmVIRILIZER, GmMTA, GmMATB, and GmMTC have three common closest orthologs of *Gossypium barbadense*, *Linum usitatissimum*, and *Anacardium occidentale*. Another writer family, GmFIP37, has the closest ortholog of *Coffea arabica* and *Solanum lycopersicum*, while *Linum usitatissimum* is the distantly related ortholog. *Zea mays* is the most distantly related ortholog of GmVIRILIZER, demonstrating the plant kingdom's evolutionary divergence. On the other hand, the red algae species *Porphyra umbilicalis* has the most distant association with the orthologs of GmMTA, suggesting a wider evolutionary range. The orthologs of GmMTB and GmMTC are most distantly linked to *Coffea arabica*, highlighting the evolutionary connection among flower species.

The YTH domain-containing eraser protein orthologs also form 4 different groups named I, II, III, and IV (Fig 2B). The various species' readers are categorized according to their class (Fig 2C). Most of these readers' orthologs of *Oryza sativa* are out-grouped from their respective clades. While monocot-monocot and dicot-dicot couples are found in certain sister taxa, monocot-dicot pairs are found in others. The closely related species *A. americanus* and *A. coerulea*, for example, show strong evidence for their nodes and a recent shared ancestor. The implication here is that monocots and dicots may have split out from a common ancestor. These three evolutionary trees demonstrate how m6A dynamics are conserved across a variety of kingdoms. The presence and functional relevance of the m6A alteration in different species within these categories have been demonstrated by these phylogenies.

## Identification of cis-regulatory elements in the putative promoter region

The spatiotemporal specific expression of protein-coding genes can be uncovered by analyzing the cis-acting regulatory elements (CREs) in the promoter region. Using the PlantCare database [52], CREs in the promoter regions (1000 bp upstream of the transcription start site) of m6A regulatory genes were predicted to better understand the transcriptional regulation and potential biological functions of soybean m6A regulatory genes (Fig 3). These CRE elements were divided into four groups, including light-responsive, phytohormone-responsive, stress-responsive, and development-related.

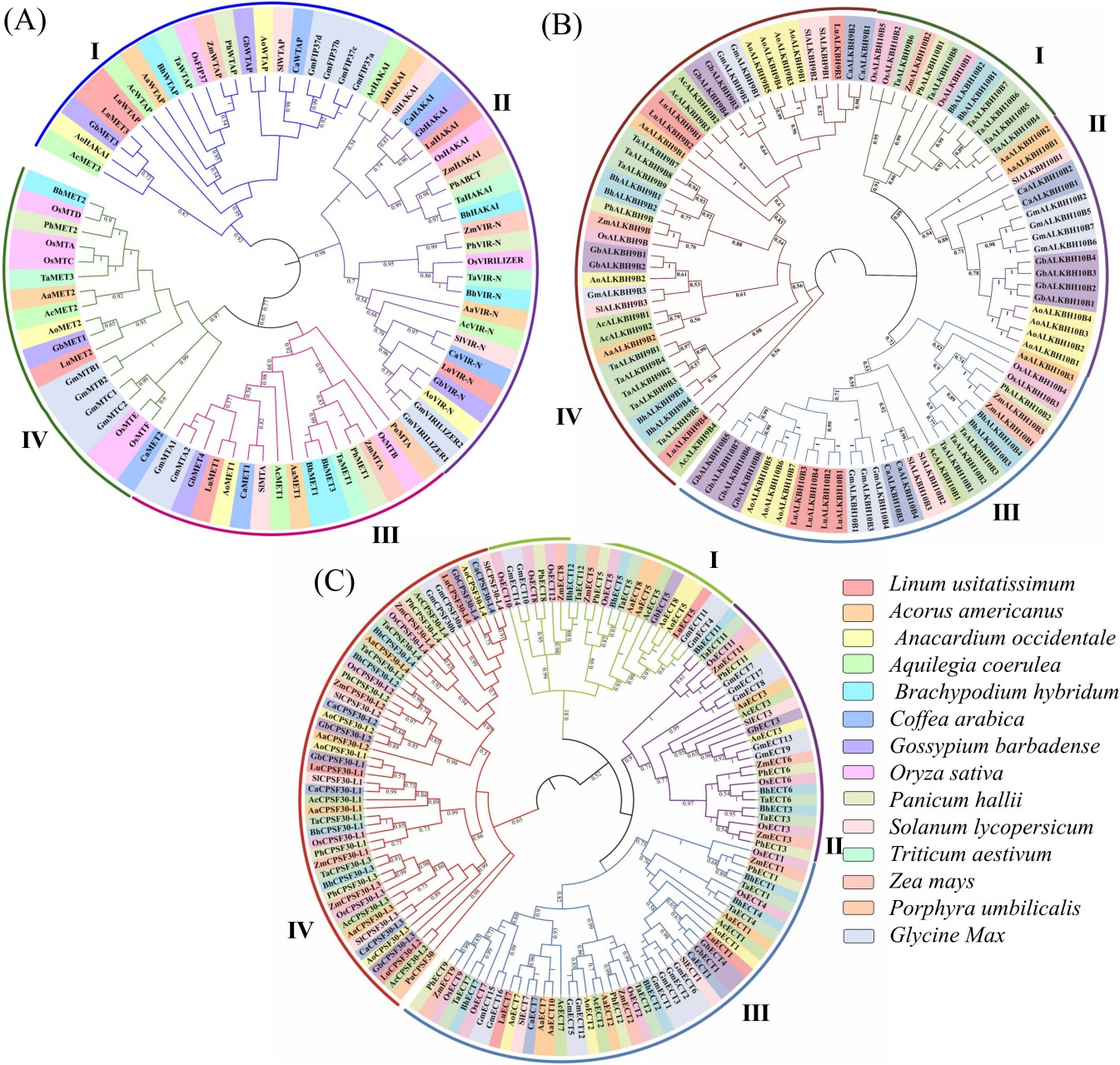

**Fig 2. Phylogenetic analysis of m6A regulatory genes from six dicotyledons, six monocotyledons, and one red alga.** A. Phylogenetic tree of writer genes. B. Phylogenetic tree of m6A eraser genes. C. Phylogenetic tree of m6A reader genes.

Light-responsive elements were predominant in all the m6A regulatory genes, suggesting that these gene sets may be regulated by light signaling. ABRE and TGA are predominant in phytohormone elements. This substantial number of phytohormone-responsive elements indicated the possible activation or inhibition of the expression level of the m6A regulatory gene by the hormone signals. In terms of development, CAT-box, GCN4-motif, ARE, and $O_2$-site were enriched in the promoter regions. These CREs were respectively associated with meristem expression, endosperm expression,

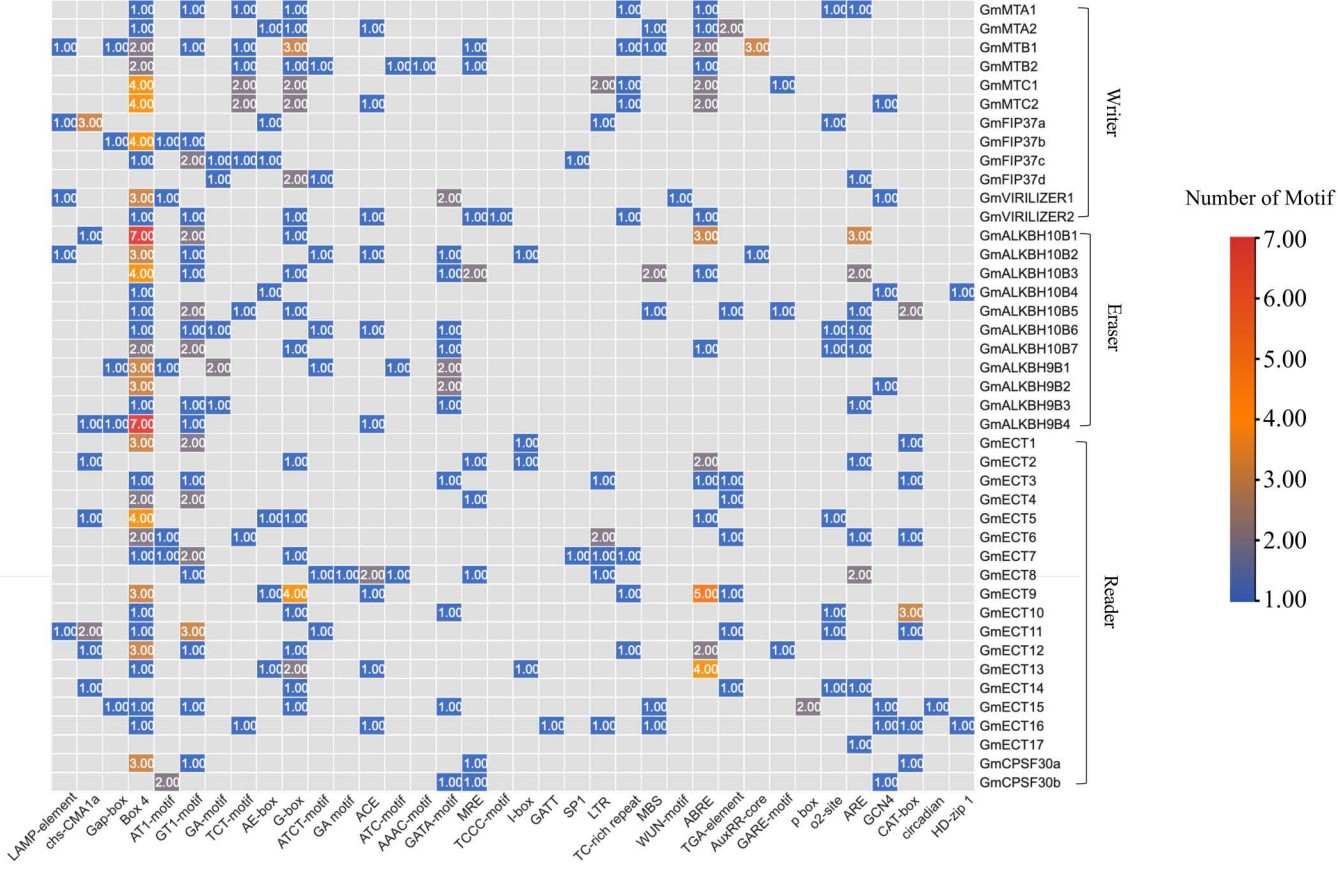

**Fig 3. The distribution of cis-acting elements in the promoter regions of m6A regulatory genes.** The 1000 bp 5′ upstream region of all the identified m6A genes was retrieved and analyzed using the PlantCARE database to determine the presence and number of cis-acting regulatory elements. The presence of selected motifs (x-axis) in the promoter of corresponding genes (Y-axis) was represented by the number of motifs in the heatmap.

anaerobic induction, and metabolism regulation. MBS (related to drought response), TC-rich repeat (related to defense and stress), and LTR (related to cold response) elements were also highly enriched.

## Functional analysis of m6A regulatory genes

The GO analysis of the m6A regulatory genes revealed that the biological process was mostly engaged in the regulation of gene expression (Fig 4A). Nucleic acid metabolic process, regulation of mRNA metabolic process, negative regulation of gene expression, and mRNA metabolic process were the dominant molecular processes. RNA binding, mRNA binding, Nucleic acid binding, and binding were observed in molecular function (Fig 4B). Intracellular anatomical structure was seen in the cellular component (Fig 4C). These results highlight the critical function of m6A regulatory genes in coordinating cellular and molecular processes that influence the dynamics of genes and RNA.

## Protein-protein interaction and cluster identification

A multi-protein complex performs together in m6A modification, highlighting the prediction of a potential protein-protein interaction network in this regard. There were 192 edges and 42 nodes in the protein-protein interaction (PPI) network (Fig 5). Two clusters were identified by using MCODE [54]. Cluster 1 contains writer proteins, including MTA, MTB,

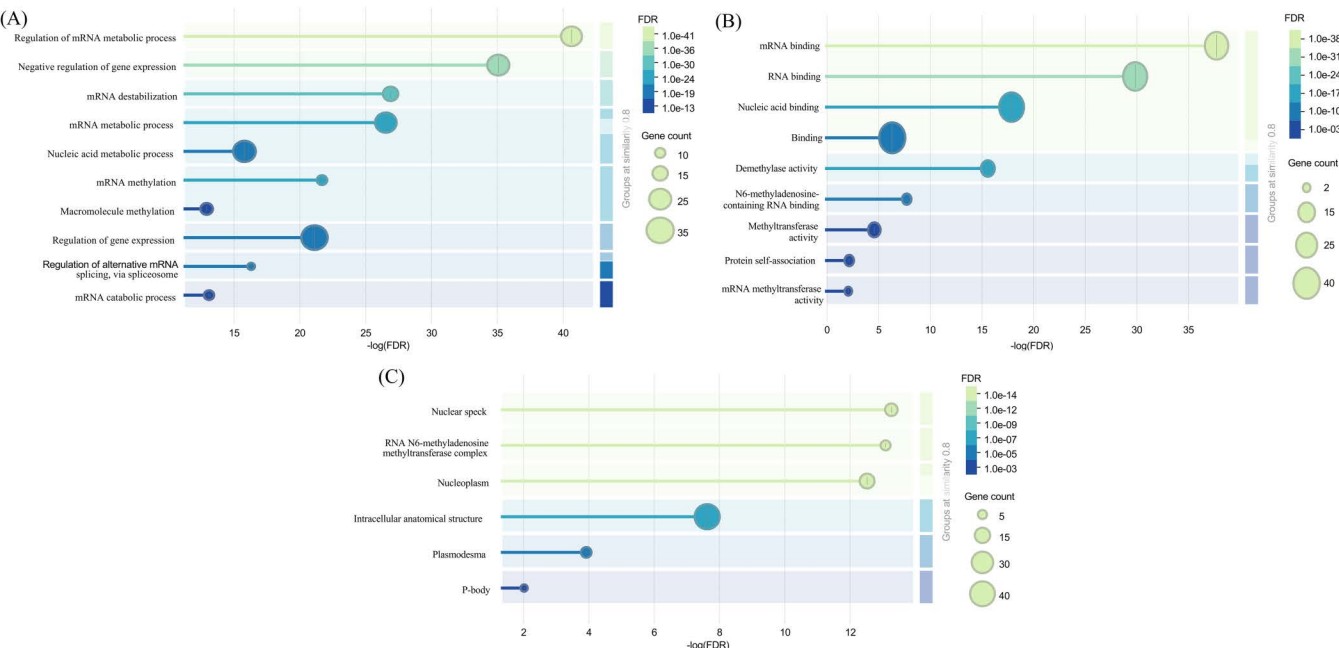

**Fig 4. Functional analysis of m6A regulatory genes in Soybean.** A. Bubble diagram illustrating the enrichment of the gene ontology for the category of biological processes. B. A bubble chart for the Gene Ontology enrichment analysis term for the category of molecular function C. Bubble diagram for the term cellular component category from the analysis of Gene Ontology enrichment. Bubble size is proportional to the number of genes, and bubble color represents the value of False Discovery Rate (FDR). Lighter green bubbles indicate lower FDR values, meaning the results are statistically significant.

and VIRILLIZER (S3A Fig). Cluster 2 contains reader and writer proteins named *GmFIP37a*, *GmFIP37b*, *GmFIP37c*, *GmFIP37d*, *MTC1*, *MTC 2*, *CPSF30a* and *ECT14* (S3B Fig). The cluster formation gives valuable insight into the biological mechanism of m6A regulatory proteins. These proteins are vital for plant growth and development, as well as for how well plants adapt to their surroundings. These findings shed light on the molecular mechanism and possible biological purposes of soybean m6A regulatory genes.

## Identification of hub genes and their 3D structure modeling

The top ten hub genes were extracted by the cytohubba plugin [55] in Cytoscape [53]. Notably, there was no eraser member in this hub gene (S3C Fig). GmVIRILIZER1 may interact with GmVIRILIZER2, GmECT10, GmECT14, GmMTB1, and GmMTB2, according to the network. GmECT10 and GmECT14 interact with GmMTB1. Furthermore, GmMTA1 engages in interactions with every hub gene in this network (S3C Fig). However, GmMTA2 has no connection with GmMTA1 or GmCPSF30a. Furthermore, GmMTB2 engages in interactions with GmMTB1, GmECT14, and GmECT10. No interaction was found between GmCPSF30a and reader hub genes. However, GmFIP37d interacts with two reader families, including GmECT10 and GmECT14. GmVIRILIZER2 interacts with GmMTB1, GmECT10, GmECT14 and GmMTB2. The protein structure of 10 hub members was predicted using SWISS-MODEL [56]. Protein structure has a polypeptide chain of amino acids specific to each protein type. The chain is folded into a secondary structure characterized by alpha-helices and beta-sheets [65] (Fig 6).

## MicroRNA target prediction

Plant microRNAs (miRNAs), small non-coding RNAs which are usually form duplexes with their targets. They play a crucial role in regulating a wide range of biological metabolic activities, including hormone signaling, plant growth, flowering

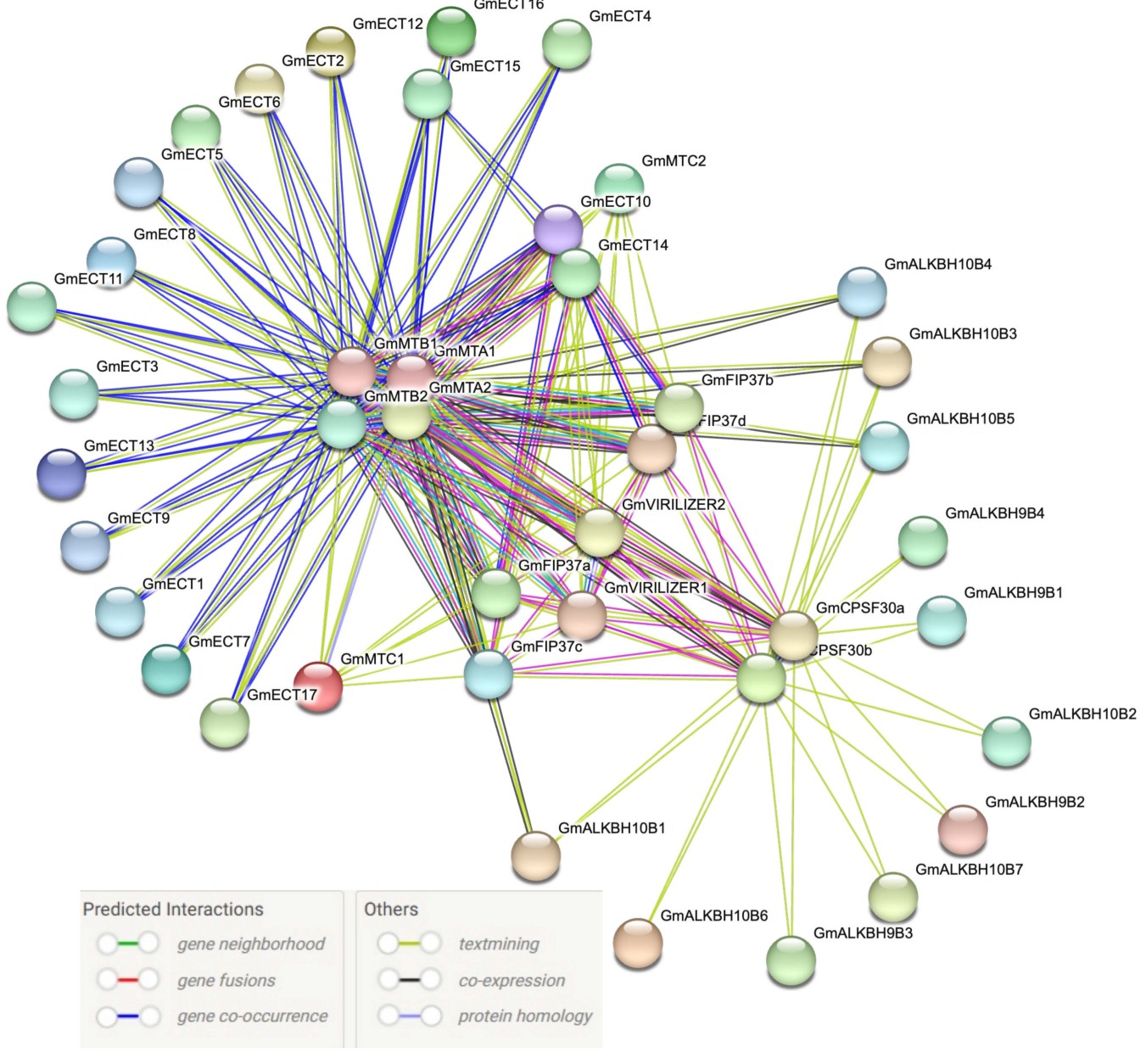

**Fig 5. PPI network of m6A regulatory genes.** PPI networks of writer eraser and reader genes were generated using the STRING database. Nodes represent the proteins, and edges represent the protein-protein associations.

time, leaf morphogenesis, and reactions to environmental stressors. Therefore, potential miRNA binding sites of m6A regulatory genes were examined in this study by using psRNATarget [59]. After filtering interactions with an Expectation penalty score below 5, it was found that 10 writers had 81 miRNAs, 10 erasers had 31 miRNAs, and 18 readers had 60 miRNAs (S4 Fig). It is noteworthy that soybean m6A writers (n = 81) seemed to be more regulated by miRNA than m6A readers (n = 60) and erasers (n = 31). Interestingly, our research did not identify any miRNAs that regulate the two writers

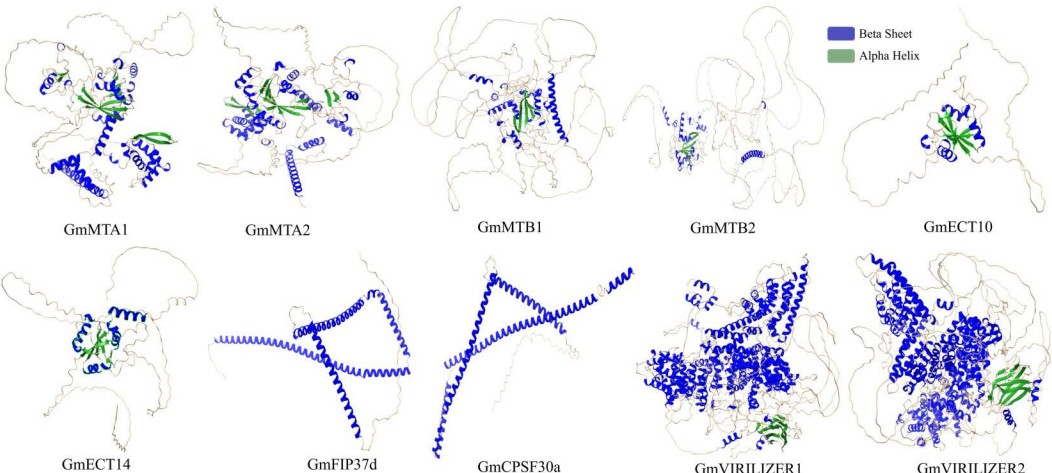

**Fig 6. 3D structure prediction of 10 hub genes using SWISS-MODEL.** Blue and green colors represent the alpha helix and beta sheets, respectively.

(*GmMTC1* and *GmMTC2*), one eraser (*ALKBH9B2*), and one reader (*ECT16*) in soybeans. However, more research is required to fully understand the complex interactions between m6A regulatory genes and miRNA.

## Expression analysis of m6A regulatory genes in roots and nodules of Soybean

Symbiotic legume nodules and lateral roots arise away from the root meristem via dedifferentiation events in soybean [61]. These organs share some morphological and developmental similarities. We explored the expression level of m6A in emerging nodules (EN), mature nodules (MN), emerging lateral roots (ELR), and young lateral roots (YLR) of soybeans. Most of the genes displayed a broad expression range across all the samples, indicating that they were extensively involved in the growth and development of roots and nodules (Fig 7). *GmMTA1* displayed vast expression levels in mature nodules. Notably, *GmMTB1*, *GmFIP37a*, *GmFIP37d*, *GmALKBH10B6*, *GmECT1*, and *GmECT12* showed a medium level of expression in all the selected samples (EN, MN, ELR, and YLR). *GmALKBH10B5* showed higher expression levels in mature nodules compared to EN, ELR, and YLR. Additionally, *GmECT9* was expressed tremendously in all the selected tissues, while *GmECT13* exhibited immense expression in EN and ELR compared to MN and YLR. Importantly, *GmECT17* showed higher expression in ELR but displayed medium expression in the other three tissues.

## Expression patterns of m6A regulatory genes under various abiotic and biotic stress

The expression pattern of m6A regulatory genes was observed under various abiotic and biotic stresses (Figs 8A and 8B). The majority of m6A regulatory genes remained unresponsive when subjected to salt stress and dehydration. *GmMTB2* and *GmECT9* exhibited significant upregulations in heat stress and combined water deficit and heat stress (Fig 8A). On the contrary, *GmALKBH9B2* showed downregulation in combined water deficit and heat stress. The remaining soybean's m6A genes displayed a medium level of expression except *GmECT4* and *GmECT2*. These two genes did not show any changes across all the abiotic stresses (Fig 8A).

The soybean mosaic virus (SMV) treatment resulted in the significantly higher up-regulation of multiple genes, such as *GmMTB1*, *GmMTB2*, and *GmALKBH9B1* (Fig 8B). In the writer's family *GmMTA2*, *GmCPSF30a*, *GmVIRILIZER1*, *GmMTA1*, *GmVIRILIZER2*, and *GmCPSF30b* showed upregulation compared to control. In the ALKBH group, three genes, such as *GmALKBH10B3*, *GmALKBH10B7*, and *GmALKBH10B6* upregulated in SMV treatment. Additionally, in

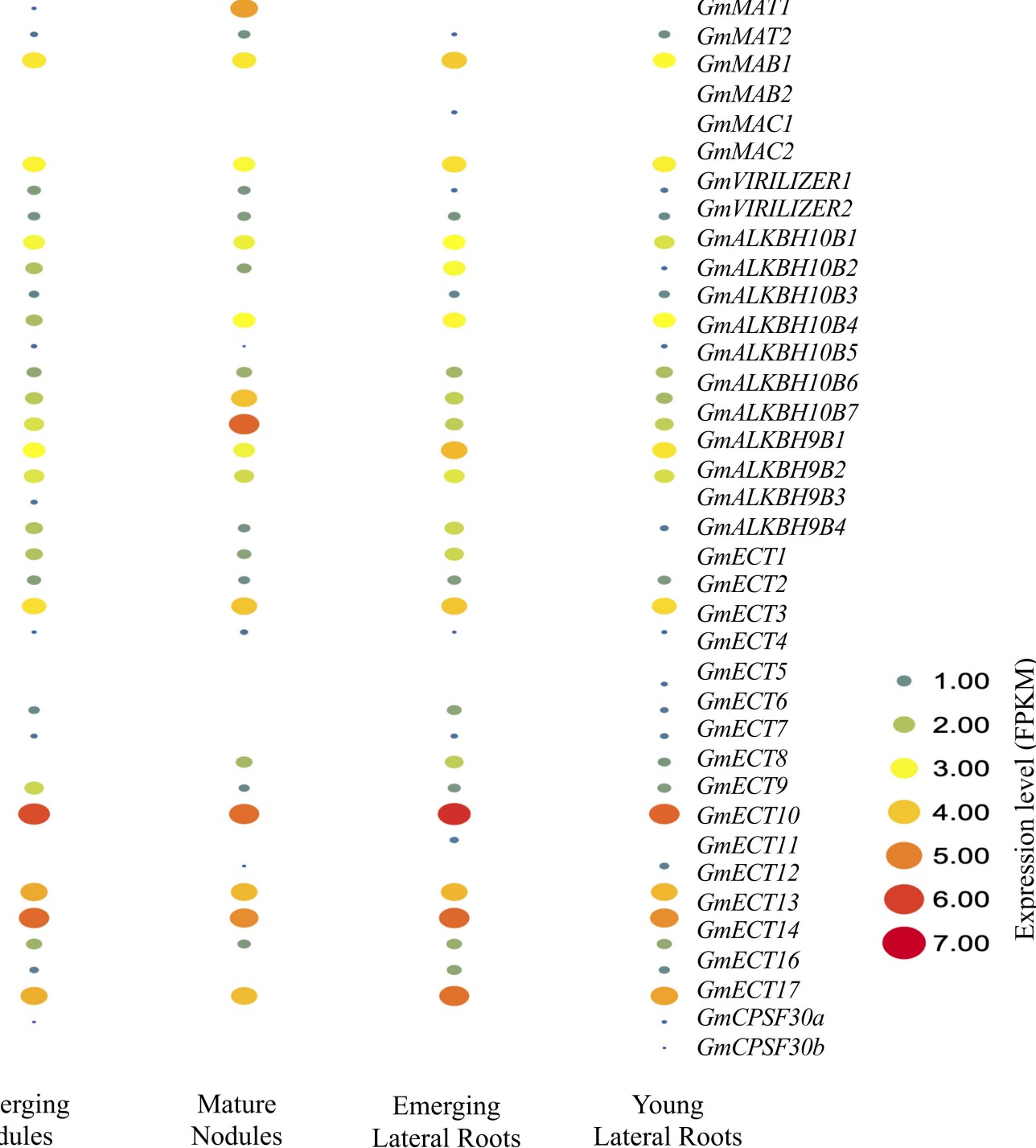

Emerging Nodules | Mature Nodules | Emerging Lateral Roots | Young Lateral Roots

**Fig 7. Expression profiling of m6A regulatory genes in roots and nodules of soybean.** Soybean RNA writer, eraser, and reader gene's genome-wide RNA-Seq data were obtained from the GEO of NCBI. Expression levels in emerging nodules, mature nodules, emerging lateral roots, and mature lateral roots were observed by using normalized FPKM (fragments per kilobase of exon per million). The color scale provided at the right of the figure represents the level of expression.

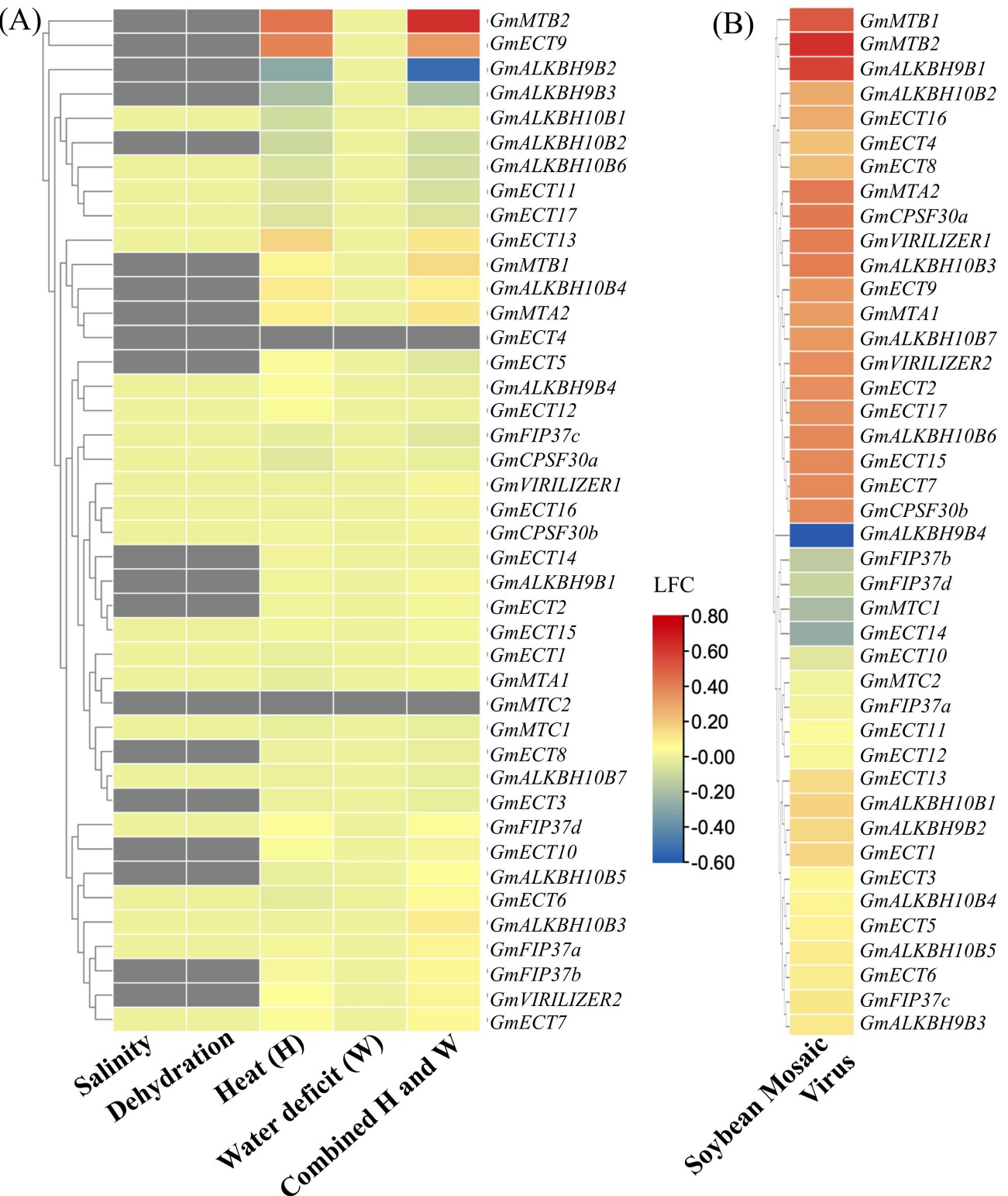

**Fig 8. Expression profiling of m6A regulatory genes under abiotic and biotic stress.** A. The expression pattern of 42 genes was analyzed in response to five abiotic stress conditions, including salt, heat, dehydration, water deficit, and combined water deficit and heat stress. B. The expression profile of writer, eraser, and reader was analyzed in response to SMV (Soybean Mosaic Virus). Heatmaps were generated using TBtool with the LFC

(Log Fold Change), which delineates the expression pattern of 42 m6A regulatory genes. The color scale provided at the right of the figure represents the level of expression. The stress-induced upregulation and downregulation of all the genes are indicated by the red and blue colors, respectively.

the readers group, *GmECT9*, *GmECT2*, *GmECT17*, *GmECT15*, and *GmECT7* displayed upregulation. *GmALKBH9B4* was the only member of the m6A regulatory gene that displayed significant downregulation in SVM treatment. Overall, the expression patterns of m6A regulatory genes in soybeans under different stress conditions indicated that they might play critical biological roles in various stress responses.

## Discussion

RNA N6-methyladenosine plays a crucial regulatory role in plant growth and development. The methylation levels of target transcripts are dynamically regulated by three types of m6A regulatory proteins. A total of 42 candidate m6A regulatory genes were identified in soybean, including 12 writers, 11 erasers, and 19 readers (Table 1). The gene sets were confirmed by the conserved domains of writer, reader, and eraser. Soybean (42) has more m6A regulatory genes than others, such as *Arabidopsis* (33), tomato (25), grape (40), rice (33), *S. moellendorffii* (22), *M. polymorpha* (16), *P. patens* (18), and Chinese pine (36) [66]. On the other hand, soybean falls short of tobacco (52) and upland cotton (75), common wheat (85), and maize (55) [66]. According to Su et al [66], a low number of m6A regulatory genes were found in Chinese pine (36) compared to species such as common wheat and upland cotton, though Chinese pine has the largest genome size (25.4 Gb). The plants that have experienced polyploidization events, such as common wheat (2n=6x=42), upland cotton (2n=4x=52), and tobacco (2n=4x=48), exhibit a higher abundance of m6A regulatory genes [67–69]. Likewise, soybean (2n=4x=40) [70], a polyploid species, also possesses a higher number of m6A regulatory genes. Additionally, maize, a segmental allopolyploid, also shows a higher number of m6A regulatory genes [71]. Using PGDD, a total of 18 duplication events were identified in this study. All the duplication events were segmental duplication (Table 2**, and** S1 Fig**).** From the Ka and Ks analysis, duplicated gene pairs went through purifying selection, a finding consistent with studies on tomatoes [72], tea plants [73], and tobacco [66].

To explore the evolutionary patterns of m6A writer, eraser, and reader genes, phylogenetic trees were constructed across 13 plant species, which included Rhodophyta (red algae) and Angiosperms (comprising eudicots and monocots) (Fig 2). Monocot-dicot, monocot-monocot, and dicot-dicot pairs were spotted in this study. For instance, *Acorus americanus* and *Aquilegia coerulea* share a recent ancestor, implying shared monocot-dicot divergence. In the previous study on m6A genes of *Oryza sativa*, a similar evolutionary pattern was observed [74]. Cis-regulatory elements in the promoter region were identified. According to this analysis, CREs were associated with light response, hormone response, plant growth and development, and stress response. Similar findings were also observed in other plant species such as tobacco [66], tea [73], tomato [72], and poplar [75]. Light-responsive CREs were predominant in soybean m6A regulatory genes (Fig 3).

GO terms in three categories, including biological process, cellular components, and molecular function, were documented. These results indicate the involvement in RNA modification of writer, eraser, and reader genes of soybean (Fig 4). The PPI network revealed that MTA1, MTA2, MTB1, and MTB2 are highly interconnected genes, indicating their importance as the catalytic core (Fig 5). Most of the erasers did not interact with the writer and the reader. On the other side, CPSF30a and CPSF30b, two readers, had stronger interaction than any other reader components (Fig 5). Cluster analysis showed two cluster formations of m6A regulatory genes of soybeans (S3 Fig). Six writer components formed cluster 1, highlighting their dynamic action in RNA modification (S3A Fig). Writer components also predominated in cluster 2; nevertheless, no erasers were found in any cluster (S3A and S3B Fig). Previous studies also found similar results that there were no interactions among eraser proteins in tobacco [66]. Furthermore, hub gene identification exhibited that there were no eraser genes in the top 10 hub genes (S3C Fig). Using homology modeling, the 3D structure of 10 hub genes

was predicted as these genes might be the core catalytic component in m6A methylation (Fig 6). The secondary structures, such as alpha helices and beta sheets, are important to maintain protein stability and functionality [76]. To explore the mutual interaction of miRNA and m6A soybean, miRNA target prediction was performed. Interestingly, the writer component had greater miRNA regulation compared to readers and erasers (S4 Fig). These findings indicate that soybean writers may prefer to be regulated by miRNA.

Researchers have demonstrated that altering the m6A system may cause irregular growth and development. Many previous studies have mentioned the association of m6A modification with reproductive cell development [77] and fruit ripening [78]. In this study, soybean m6A expression was observed in roots and nodules. Readers and erasers have a higher expression than writers regarding roots and nodules (Fig 7). Two reader genes named *GmECT9* and *GmECT13* are highly expressed in the respective samples. According to prior studies, *CPSF30L* in Arabidopsis might regulate APA by binding to m6A-modified RNAs that are involved in nitrate signaling, particularly WRKY1 and NRT1.1 [79]. As *GmECT9* and *GmECT13* were highly expressed in roots and nodules, it can be hypothesized that these genes have a role in nitrogen absorption. However, further investigation is needed in this regard.

CRE elements such as MBS (related to drought response), TC-rich repeat (related to defense and stress), and LTR (related to cold response) were highly enriched in soybean m6A genes according to Promoter analysis (Fig 3). This result indicates the involvement of soybeans' m6A genes in environmental stimuli. Salt stress has been shown to have a major impact on the m6A methylation levels of mRNA in Arabidopsis [35]. The expression patterns of *GmMTA*s and *GmMTB*s under various abiotic stressors have been documented in earlier research, suggesting their possible role in stress tolerance, particularly in the reaction to darkness or alkalinity [39]. According to another research, *GmALKB10B* genes were probably triggered by alkalinity, cold, and dryness [40]. In this study, five abiotic stress responses, such as salt stress, dehydration, heat stress, water deficit, and combined water and heat have been reported. Most of the soybean's m6A genes did not show any changes in salt and dehydration stress, indicating their stress tolerance possibility (Fig 8A). Notably, two genes named *GmMTB2* and *GmALKBH9B2* showed upregulation and downregulation, respectively, in combined water and heat stresses. Interestingly, *GmECT4* and *GmECT2* did not show any changes across all the abiotic stresses. Therefore, we assumed that these two genes might be stress tolerance m6A genes in soybeans. Validating these findings requires a wet lab inquiry. Distinct biotic stresses also have a great impact on the plant life cycle. *NtALKBH10* and *NtVIR1* of tobacco exhibited robust responses to specific biotic stresses (R. solanacearum and black shank) but showed insensitivity to the Cucumber Mosaic Virus (CMV) [66]. In this study, biotic stress responses (soybean mosaic virus) have been documented. Most of the soybean's m6A genes displayed upregulation in SMV treatment (Fig 8B). Nevertheless, one member called *GmALKBH9B4 was* downregulated in this regard. In conclusion, the m6A regulatory genes of soybeans showed a variety of expression patterns under stress, nodules, and roots, suggesting that these genes have a multitude of roles. The precise processes of the putative m6A regulatory genes in soybeans require further investigation.

## Conclusion

This study presents a comprehensive and systematic investigation of m6A regulatory genes in soybean, and a total of 42 m6A regulatory genes were identified. We analyzed the features of these genes in terms of gene structure, conserved domains, and motifs. Phylogenetic study and duplication events were also examined. Eighteen duplication events that evolved through purifying selection were found. Functional analysis of m6A regulatory genes in cis-elements and interaction networks demonstrated their crucial roles in soybeans. The altered expression pattern of these genes had been profiled in roots, nodules, and under distinct stresses. For instance, the three readers—*GmECT9*, *GmECT13*, and *GmECT17*—were highly expressed in roots and nodules. *GmMTB2* exhibited the highest upregulation in response to combined water deficit and heat stress, while *GmALKBH9B2* showed the greatest downregulation. Furthermore, *GmALKBH9B4* was significantly downregulated, whereas *GmMTB1*, *GmMTB2*, and *GmALKBH9B1* were upregulated under

soybean mosaic virus treatment. This study gives a reference framework for investigating the functional variety of soybean m6A regulatory genes at the epigenetic level.

## Supporting information

**S1 Fig. Chromosomal distribution of m6A regulatory genes.** All m6A regulatory genes are found to be located on different chromosomes of soybeans. The relative size of the corresponding chromosomes and the position of the respective genes could be estimated by using the scale provided left side of the figure. The chromosome numbers are provided in the middle of each bar. The straight lines connect the duplicated gene pairs.
(PDF)

**S2 Fig. Organization and distribution of the conserved motifs.** The organization and distribution of the conserved motifs in the m6A writer (A), eraser (B), and reader (C) genes. The squares in the motif represent the positions of conserved domains.
(PDF)

**S3 Fig. Cluster and hub genes identification in soybeans.** Two clusters (A and B) were identified using MCODE. C. The top 10 hub genes were extracted by the Density of Maximum Neighborhood Component method in the cytohubba plugin in Cytoscape.
(PDF)

**S4 Fig. Genome-wide network of soybean miRNA.** Genome-wide miRNA-regulated networks of the writer (A), eraser (B), and reader (C). Green nodes: miRNAs, Pink nodes: genes that may be miRNA targets, and Black edges: correlations.
(PDF)

## Acknowledgments

The authors extend their gratitude to the Department of Biochemistry and Molecular Biology at Shahjalal University of Science and Technology, Sylhet, Bangladesh, for providing lab facilities and logistic support.

## Author contributions

**Conceptualization:** Ajit Ghosh.

**Data curation:** Sabrina Bintay Sayed.

**Formal analysis:** Sabrina Bintay Sayed, Md. Afser Rabbi.

**Investigation:** Sabrina Bintay Sayed, Joy Prokash Debnath, Kabir Hossen.

**Methodology:** Sabrina Bintay Sayed, Md. Afser Rabbi, Joy Prokash Debnath, Kabir Hossen.

**Project administration:** Ajit Ghosh.

**Supervision:** Ajit Ghosh.

**Validation:** Ajit Ghosh.

**Writing – original draft:** Sabrina Bintay Sayed.

**Writing – review & editing:** Ajit Ghosh.

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
