## [Decision Letter · Decision Letter 0]

PONE-D-25-28327

Genome-Wide Identification and Characterization of m6A Regulatory Genes in Soybean: Insights into Evolution, miRNA Interactions, and Stress Responses

PLOS ONE

Dear Dr. Ghosh,

Thank you for submitting your manuscript to PLOS ONE. After careful consideration, we feel that it has merit but does not fully meet PLOS ONE’s publication criteria as it currently stands. Therefore, we invite you to submit a revised version of the manuscript that addresses the points raised during the review process.

We look forward to receiving your revised manuscript.

Kind regards,

Kai Huang

Academic Editor

PLOS ONE

Journal Requirements:

https://bmcplantbiol.biomedcentral.com/articles/10.1186/s12870-024-04813-2

In your revision ensure you cite all your sources (including your own works), and quote or rephrase any duplicated text outside the methods section. Further consideration is dependent on these concerns being addressed.

3. Please include your tables as part of your main manuscript and remove the individual files. Please note that supplementary tables (should remain/ be uploaded) as separate "supporting information" files.

Reviewers' comments:

Reviewer's Responses to Questions

**Comments to the Author**

1. Is the manuscript technically sound, and do the data support the conclusions?

Reviewer #1: Yes

Reviewer #2: Yes

2. Has the statistical analysis been performed appropriately and rigorously? 

Reviewer #1: No

Reviewer #2: Yes

3. Have the authors made all data underlying the findings in their manuscript fully available?

Reviewer #1: Yes

Reviewer #2: Yes

4. Is the manuscript presented in an intelligible fashion and written in standard English?

Reviewer #1: Yes

Reviewer #2: Yes

5. Review Comments to the Author

Reviewer #1: In this article, the authors report the identification of genes encoding 12 m6A writer, 11 eraser and 19 reader proteins in soybean. They characterized the gene and domain structures, and their interactomes. They also showed that expression of these genes might be controlled by light exposure, stress, or plant growth. Overall, they authors provided the first characterization of m6A effector proteins in soybean highlighting potential key role in plant development, especially when experiencing stress. Here are some comments I would like to see addressed:

1) Table 1 and 2 were not included in the manuscript. These are key to the manuscript, and therefore I was not able to thoroughly review this manuscript. Please ensure they are included.

2) Please discuss previous literature results from reference 40. They identified 13 writers, including two HAKAI-containing ones. Can you comment on the differences? Are there HAKAI-containing writers in soybean? Please double check your results and add clear discussion of the reference results. Also modify lines 414-415 accordingly.

3) Figure 7 and 8: Please add significance evaluation for expression levels. Are these differences significant?

4) Lines 365-373: Please add rationale for looking at miRNA target

5) All Figures: When downloaded the quality was good, however when added to the PDF the quality prevented clear reading. Ensure the sizing of your figure matches the final size in the article to prevent reduced quality.

6) Figure 1: Ensure the color-coding of the legend matches the figure

7) Figure 3: Please label legend in the figure: what is the scale showing?

8) Figure 4: Do the color of the gene bubbles represent something? Please clarify.

9) Figure 8: Please label legends in the figure.

10) Lines 330-332: Please remove unnecessary capital letters.

11) Lines 411-412: “are regulated by three types of m6A regulatory genes” is incorrect. Methylation is regulated by proteins, or gene expression but not gene themselves.

12) Line 455: Typo “sheath” should be “sheets”

13) Line 482: “combines” Please removed unnecessary capital letter.

Reviewer #2: This manuscript presents a comprehensive and well-structured genome-wide analysis of m6A regulatory genes in soybean, including writers, erasers, and readers. Through an integrative bioinformatic pipeline, the authors examine gene structure, evolutionary relationships, duplication events, protein-protein interactions, miRNA targeting, and expression profiles under abiotic and biotic stress conditions.

The work addresses a relevant knowledge gap in plant epitranscriptomics, particularly in soybean, a crop of major agricultural importance. The study is timely, methodologically sound, and provides valuable data that will be of interest to the plant biology and epigenetics communities.

6. PLOS authors have the option to publish the peer review history of their article (what does this mean? ). If published, this will include your full peer review and any attached files.

**Do you want your identity to be public for this peer review?** For information about this choice, including consent withdrawal, please see our Privacy Policy .

Reviewer #1: No

Reviewer #2: No

---

## [Author Response · Author response to Decision Letter 1]

26 Jun 2025

Comments from the Scientific Handling Editor:

Comment 1: “Please ensure that your manuscript meets PLOS ONE's style requirements, including those for file naming. The PLOS ONE style templates can be found at

https://journals.plos.org/plosone/s/file?id=wjVg/PLOSOne_formatting_sample_main_body.pdf and https://journals.plos.org/plosone/s/file?id=ba62/PLOSOne_formatting_sample_title_authors_affiliations.pdf”

Response: We have revised our manuscript to fully comply with PLOS ONE’s formatting guidelines, including file naming and structure, as per the provided templates.

Comment 2: “We noticed you have some minor occurrence of overlapping text with the following previous publication(s), which needs to be addressed:

https://bmcplantbiol.biomedcentral.com/articles/10.1186/s12870-024-04813-2

In your revision ensure you cite all your sources (including your own works), and quote or rephrase any duplicated text outside the methods section. Further consideration is dependent on these concerns being addressed.”

Response: We have carefully reviewed the manuscript for any overlapping content with the cited publication. All such instances outside the Methods section have been rephrased appropriately.

Comment 3: “Please include your tables as part of your main manuscript and remove the individual files. Please note that supplementary tables (should remain/ be uploaded) as separate "supporting information" files.”

Response: All main tables have now been incorporated into the main manuscript file.

Comment 4: “Please include captions for your Supporting Information files at the end of your manuscript, and update any in-text citations to match accordingly. Please see our Supporting Information guidelines for more information: http://journals.plos.org/plosone/s/supporting-information.”

Response: Captions for all Supporting Information files have been added at the end of the revised manuscript.

Response to Reviewer #1

Comment 1: “Table 1 and 2 were not included in the manuscript. These are key to the manuscript, and therefore I was not able to thoroughly review this manuscript. Please ensure they are included.”

Response: We apologize for the oversight. Tables 1 and 2 have now been included in the revised manuscript.

Comment 2: “Please discuss previous literature results from reference 40. They identified 13 writers, including two HAKAI-containing ones. Can you comment on the differences? Are there HAKAI-containing writers in soybean? Please double check your results and add clear discussion of the reference results. Also modify lines 414-415 accordingly”.

Response: Thank you for the valuable suggestion. Based on phytozome data, we also identified a HAKAI-related protein (GmHAKAI) in soybean. However, Pfam domain analysis did not confirm the presence of a HAKAI domain. Consequently, GmHAKAI was excluded from downstream analysis. We mentioned it in the result section and have modified lines 411-415 accordingly.

Comment 3: “Figure 7 and 8: Please add significance evaluation for expression levels. Are these differences significant?”

Response: In Figure 7, we used normalized FPKM values to evaluate the expression levels of m6A regulatory genes in emerging nodules (EN), mature nodules (MN), emerging lateral roots (ELR), and young lateral roots (YLR) of soybean. While variation in expression was observed across the tissues, genes such as GmECT9 and GmECT13 exhibited relatively higher expression compared to others. In Figure 8A, we extracted LFC (Log Fold Change) using DESec2. Some genes remain unresponsive when subjected to abiotic stress, which might hint stress tolerance feature of those genes. We also examined the expression pattern in biotic stress, such as Soybean Mosaic Virus treatment. In Figure 8B, GmALKBH9B4 was the only gene that displayed significant downregulation in SVM treatment. We have now revised both figures and legends to indicate statistically significant differences.

Comment 4: “Lines 365-373: Please add rationale for looking at miRNA target.”

Response: The rationale for analyzing miRNA targets has now been added to the manuscript in the relevant section (Lines 365–373).

Comment 5: “All Figures: When downloaded, the quality was good, however when added to the PDF the quality prevented clear reading. Ensure the sizing of your figure matches the final size in the article to prevent reduced quality”.

Response: Thank you for bringing this to our attention. All figures have been resized and optimized for print resolution to ensure clarity in the final PDF version.

Comment 6: “Figure 1: Ensure the color-coding of the legend matches the figure.”

Response: We have carefully reviewed Figure 1 and corrected the legend to ensure that it accurately reflects the color coding used in the figure.

Comment 7: “Figure 3: Please label the legend in the figure: what is the scale showing?”

Response: We have added a clear legend to Figure 3 indicating that the scale represents the number of motifs.

Comment 8: “Figure 4: Do the color of the gene bubbles represent something? Please clarify”.

Response: The color of the gene bubble represents the FDR (False Discovery Rate). Lower FDR has greater significance. Yes, the color of the gene bubbles in Figure 4 represents the False Discovery Rate (FDR), where lower FDR values indicate higher statistical significance. This explanation has now been added to the figure legend.

Comment 9: “Figure 8: Please label legends in the figure”.

Response: Legends have been added to Figure 8 to clarify all graphical elements.

Comment 10: “Lines 330-332: Please remove unnecessary capital letters.”

Response: The unnecessary capitalizations have been removed.

Comment 11: “Lines 411-412: “are regulated by three types of m6A regulatory genes” is incorrect. Methylation is regulated by proteins, or gene expression but not gene themselves.”

Response: Thank you for this clarification. The sentence has been corrected to reflect that methylation is regulated by m6A-related proteins, not the genes themselves.

Comment 12: “Line 455: Typo ‘sheath’ should be ‘sheets’”

Response: The typographical error has been corrected.

Comment 13: “Line 482: “combines” Please remove unnecessary capital letter.”

Response: The capitalization error has been corrected.

Response to Reviewer #2

“This manuscript presents a comprehensive and well-structured genome-wide analysis of m6A regulatory genes in soybean, including writers, erasers, and readers. Through an integrative bioinformatic pipeline, the authors examine gene structure, evolutionary relationships, duplication events, protein-protein interactions, miRNA targeting, and expression profiles under abiotic and biotic stress conditions.

The work addresses a relevant knowledge gap in plant epi-transcriptomics, particularly in soybean, a crop of major agricultural importance. The study is timely, methodologically sound, and provides valuable data that will be of interest to the plant biology and epigenetics communities.”

Response: We sincerely thank Reviewer #2 for the positive and encouraging comments regarding the quality and significance of our work. This thoughtful review reinforces the value of our integrative bioinformatics approach and motivates us to continue contributing to this important field of research.

---

## [Decision Letter · Decision Letter 1]

Genome-wide identification and characterization of m6A regulatory genes in Soybean: Insights into evolution, miRNA interactions, and stress responses

PONE-D-25-28327R1

Dear Dr. Ghosh,

We’re pleased to inform you that your manuscript has been judged scientifically suitable for publication and will be formally accepted for publication once it meets all outstanding technical requirements.

Kind regards,

Kai Huang

Academic Editor

PLOS ONE

Additional Editor Comments (optional):

Reviewers' comments:

Reviewer's Responses to Questions

**Comments to the Author**

1. If the authors have adequately addressed your comments raised in a previous round of review and you feel that this manuscript is now acceptable for publication, you may indicate that here to bypass the “Comments to the Author” section, enter your conflict of interest statement in the “Confidential to Editor” section, and submit your "Accept" recommendation.

Reviewer #1: All comments have been addressed

Reviewer #2: All comments have been addressed

2. Is the manuscript technically sound, and do the data support the conclusions?

Reviewer #1: (No Response)

Reviewer #2: Yes

3. Has the statistical analysis been performed appropriately and rigorously? 

Reviewer #1: (No Response)

Reviewer #2: Yes

4. Have the authors made all data underlying the findings in their manuscript fully available?

Reviewer #1: (No Response)

Reviewer #2: Yes

5. Is the manuscript presented in an intelligible fashion and written in standard English?

Reviewer #1: (No Response)

Reviewer #2: Yes

6. Review Comments to the Author

Reviewer #1: (No Response)

Reviewer #2: The revised manuscript presents a comprehensive genome-wide identification and characterization of m6A regulatory genes in soybean, integrating evolutionary analysis, domain architecture, gene duplication, miRNA targeting, and stress-responsive expression profiling. The authors have provided detailed responses to all prior reviewer and editor comments and have appropriately revised the manuscript to address the raised concerns.

7. PLOS authors have the option to publish the peer review history of their article (what does this mean? ). If published, this will include your full peer review and any attached files.

**Do you want your identity to be public for this peer review?** For information about this choice, including consent withdrawal, please see our Privacy Policy .

Reviewer #1: No

Reviewer #2: **Yes: ** Haozhou Tan

---

## [Editor Report · Acceptance letter]

PONE-D-25-28327R1

PLOS ONE

Dear Dr. Ghosh,

I'm pleased to inform you that your manuscript has been deemed suitable for publication in PLOS ONE. Congratulations! Your manuscript is now being handed over to our production team.

Kind regards,

on behalf of

Dr. Kai Huang

Academic Editor

PLOS ONE